# The Effect of Aerobic Exercise on Variation of Oxidative Stress, hs-CRP and Cortisol Induced by Sleep Deficiency

**DOI:** 10.3390/healthcare11081201

**Published:** 2023-04-21

**Authors:** Jong-Suk Park, Zsolt Murlasits, Sangho Kim

**Affiliations:** 1School of Global Sport Studies, Korea University, Sejong-si 30019, Republic of Korea; model200@korea.ac.kr; 2Institute of Sport Sciences and Physical Education, Faculty of Sciences, University of Pécs, 7624 Pécs, Hungary; drmurla@gamma.ttk.pte.hu

**Keywords:** sleep deficiency, aerobic exercise, oxidative stress, hs-CRP, cortisol

## Abstract

The aim of this study was to investigate the impact of sleep deficiency (SD) on oxidative stress, hs-CRP and cortisol levels and to examine the effects of different intensities of aerobic exercise on these parameters under SD conditions. Thirty-two healthy male university students participated in the study and underwent both normal sleep (NS, 8 h of sleep per night for 3 consecutive days) and SD (4 h of sleep per night for 3 consecutive days). After the SD period, the participants performed treatment for 30 min according to their assigned group [sleep supplement after SD (SSD), low-intensity aerobic exercise after SD (LES), moderate-intensity aerobic exercise after SD (MES), high-intensity aerobic exercise after SD (HES)]. Sleep-related factors were measured at NS and SD, while oxidative stress, hs-CRP and cortisol levels were measured at NS, SD and immediately after treatment by group (AT). The results showed that actual total sleep time (ATST) was significantly reduced during SD compared to NS (*p* < 0.001), while the visual analogue scale (VAS) and Epworth sleepiness scale (ESS) were significantly increased during SD compared to NS (*p* < 0.001). The difference in reactive oxygen metabolites (d-ROMs) and cortisol levels showed a significant interaction effect (*p* < 0.01, *p* < 0.001, respectively), with LES showing a decrease in d-ROMs and cortisol levels compared to SD (*p* < 0.05). Similarly, SSD showed a decrease in cortisol levels compared to SD (*p* < 0.05), while HES led to a significant increase in d-ROMs and cortisol levels compared to SD (*p* < 0.05). Biological antioxidant potential (BAP) and hs-CRP did not show any significant effect (*p* > 0.05). These results suggest that LES is the most effective exercise intensity for mitigating the negative effects of SD.

## 1. Introduction

Sleep is a biological activity that is necessary for survival and well-being. It plays a vital role in brain function and systemic physiology, such as metabolism, a function of immunological, hormonal and cardiovascular systems [1]. The quantity and quality of sleep are critical factors in physical performance and well-being [2,3]. Although sleep is a critical aspect of human health, sleep deficiency (SD) has become a prevalent problem in modern society. The proportion of people who experience SD has been rapidly increasing in both the United States and South Korea [4,5].

In humans, SD has been associated with various detrimental health effects. SD has been linked to increased activity of the hypothalamic–pituitary–adrenal (HPA) axis, resulting in elevated levels of cortisol and catecholamines, as well as increased oxidative stress, high sensitivity C-reactive protein (hs-CRP) and immune cytokines [6,7,8]. SD can also impact metabolism and the immune system, leading to a higher risk of diseases such as obesity, type 2 diabetes, hypertension, cardiovascular disease and higher mortality rates [9,10,11]. Therefore, it is important to find effective interventions to prevent and mitigate the negative effects of SD on health and well-being. 

Regular exercise has been widely recommended as an intervention method for chronic diseases, such as obesity, metabolic syndrome, type 2 diabetes and cardiovascular disease [12,13]. Previous studies have shown that regular aerobic exercise can decrease blood pressure, reduce the risk of cardiovascular diseases and improve blood lipid levels [14,15]. Additionally, moderate endurance exercise can enhance immune function [16]. People with SD engage in exercise for leisure and to maintain their health because of its health benefits. However, the effects of aerobic exercise on people with SD have not been extensively studied.

Previous studies have reported that exercise may alleviate health problems associated with SD [17,18,19]. For instance, one study found that moderate exercise can prevent memory impairment resulting from SD [17]. Another study reported that aerobic exercise performed under SD conditions can reduce metabolites that cause insulin resistance and increase those that alleviate depressive symptoms caused by SD [18]. Moreover, physical activity can have beneficial effects on factors associated with obesity in people with SD [19]. However, people with sleep problems should be cautions when exercising because epidemiologic studies suggest that they have a higher risk of developing coronary heart disease, sudden cardiac death, myocardial infarction, stroke and angina [20,21]. Therefore, people with SD should be careful when applying the same exercise intensity as healthy people due to their higher risk of health complications. Despite this, the efficiency and safety of aerobic exercise as an intervention for health problems caused by SD have not been thoroughly investigated, and the most effective exercise intensity for optimal health benefits under SD conditions remains unclear. 

Most sleep-related studies have focused on complete sleep deprivation lasting 24 h. However, in reality, partial sleep restriction, such as SD, is more common and may have different effects on health and performance. Therefore, future studies should investigate the effects of partial sleep restriction on various aspects of health and functioning, as well as the potential benefits of interventions, such as exercise, in mitigating the negative effects of SD.

Therefore, this study aims to examine the negative effects of SD on oxidative stress, hs-CRP and cortisol levels and to investigate the potential effects of aerobic exercise in reducing these the negative impacts. Through this study, we aim to evaluate the efficacy and safety of aerobic exercise under SD conditions.

## 2. Materials and Methods

### 2.1. Participants

The study’s sample size was determined using G*Power software (version 3.1.9.2), with a power (1-β) of 0.95, a significance level (α) of 0.05 and an effect size of 0.4. This resulted in a calculated sample size of 28 participants. However, to account for a potential dropout rate of 10%, 32 healthy male university students with a normal body mass index (BMI) who did not exercise regularly were recruited. Participants were excluded from the study if they (1) reported a habitual sleep duration of less than 7 h, (2) had an irregular sleep–wake schedule, (3) were diagnosed with a circadian or sleep disorder, (4) were smokers, (5) exercised regularly, (6) had musculoskeletal or neurological disease or (7) had metabolic, cardiovascular or chronic inflammatory disease. All participants were informed about the purpose and procedures of the study and provided written consent. Participants were randomly assigned to one of four groups: the sleep supplement after SD (SSD, *n* = 8), the low-intensity aerobic exercise after SD (LES, *n* = 8), the moderate-intensity aerobic exercise after SD (MES, *n* = 8) and the high-intensity aerobic exercise after SD (HES, *n* = 8) (Figure 1).

The study was approved by the Institutional Review Board of Korea University (1040548-KU-IRB-16-286-A-1) before the experiment and was conducted by their guidelines and the Declaration of Helsinki (1964). Additionally, the study was registered with the Clinical Research Information Service (CRIS), a public trial registry of the Republic of Korea (CRIS-KCT0006684). 

### 2.2. Experimental Procedure

The experimental procedure of this study is illustrated in Figure 2. Initially, to determine the exercise intensity of the participants, maximal oxygen uptake (VO_2_ max) was measured using a graded treadmill exercise test one week before the normal sleep (NS) period. Subsequently, all participants underwent two different sleep conditions. Referring to previous studies [22,23], the first sleep condition was NS, with 8 h of sleep per night for 3 consecutive nights (from 11 p.m. to 7 a.m.). The second sleep condition was SD, with only 4 h of sleep per night for 3 consecutive nights (from 3 a.m. to 7 a.m.). To maintain consistency, the researchers provided the participants with the same daily diet consisting of 3 meals of 900 kcal each and prohibited any consumption except water after dinner, as well as prohibiting exercise during the experimental periods. The participants’ sleep was conducted in their own homes to avoid any disturbances due to an unfamiliar environment. The sleep environment was kept dark and quiet by turning off the lights during sleeping hours, and the researchers supervised the participants using mobile messages to ensure adherence to the experimental protocols. Following the SD periods, participants underwent a 30-min treatment according to their assigned group. Blood samples were collected from participants three times for blood analysis: after normal sleep, after sleep deficiency and immediately after the group treatment (AT). 

### 2.3. Treadmill Test

In the study, the VO_2_ max of each participant was measured before the NS period to determine the exercise intensity for the exercise groups. To minimize variability, the measurement was performed at the same time (9:00~11:00), and participants were required to fast overnight and abstain from alcohol consumption the day before the experiment. Participants performed a maximal exercise test using a treadmill (Cosmed T150, h/p Cosmos, Nussdorf-Traunstein, Germany) and an autonomous respiratory gas analyzer (TrueOne 2400, ParvoMedics, Inc., Salt Lake City, UT, USA). The Bruce protocol was used as the loading method, which involved increasing both speed and grade at 3-min intervals. The Bruce protocol used in this study is considered an appropriate exercise protocol for healthy adults. 

During the exercise, the researcher encouraged the participants to reach their maximum exercise capacity. The researchers monitored the participants’ heart rate (HR) and the results of the gas analysis, including ventilation, oxygen consumption, respiration exchange ratio (RER) and respiratory rate. This study determined VO_2_ max using objective criteria, including an HR greater than 90% of HRmax (calculated as 220 minus the participant’s age), a plateau in the oxygen uptake curve, an RER greater than 1.15 during the test and the participant’s report of fatigue. All tests were performed in a room maintained at constant temperature (23~24 °C) and relative humidity (50~55%).

### 2.4. Measurement of Sleep-Related Factors

The participants in the study were provided with a wearable activity tracker (Fitbit Charge 2, Fitbit Inc., USA) to measure their sleep and daily activity levels during the experiment period. The tracker was worn on the participant’s non-dominant wrist and equipped with a built-in triaxial accelerometer, altimeter, heart rate monitor and vibration motor. Fitbit devices are known to have a high accuracy rate of 98% when identifying sleep stages compared to polysomnography and have an inter-device reliability greater than 96% [24]. The actual total sleep time (ATST), sleep time of rapid eye movement (REM), sleep time of light sleep, sleep time of deep sleep, daily activity levels and calorie expenditure during each sleep period were recorded. The mean values of these variables over three days for each sleep period were used for analysis. In addition, we calculated the proportion of REM sleep, light sleep and deep sleep relative to the ATST.

The sleep quality, daytime sleepiness and fatigue of the participants during the experiment period were assessed using the Pittsburgh Sleep Quality Index (PSQI), the Epworth Sleepiness Scale (ESS) and the Visual Analogue Scale (VAS). The PSQI is a 19-item self-rated questionnaire that measures subjective sleep quality. The questions are divided into seven components, with each component scored on a scale of 0 to 3, where higher scores indicate poorer sleep quality. The PSQI score is the sum of the component scores, with a range of 0–21. The ESS is a widely used scale that measures sleepiness and consists of 8 self-rated items, each scored from 0 to 3, that assess a participant’s likelihood of dozing off in various everyday situations. The ESS score is calculated as the sum of the individual item scores, with a range of 0–24. The VAS is used to measure subjective fatigue levels and consists of a 10-cm line with endpoints labeled “I do not feel tired.” and “I feel extremely tired, exhausted.” Participants marked a point on the line that corresponded to their level of fatigue, and the VAS score was calculated as the distance in centimeters from the marked point to the left end of the line. The score ranged from 0 to 10, where higher scores indicate higher levels of fatigue. The PSQI was measured during the recruitment, while the ESS and VAS were measured at three different time points: during recruitment, after three days of NS and after three days of SD.

### 2.5. Treatment of Groups

Based on the participants’ VO_2_ max, they performed aerobic exercise on a treadmill for 30 min according to their assigned groups [LES (40% of VO_2_ max), MES (60% of VO_2_ max) or HES (80% of VO_2_ max)] in the morning after the SD period. The exercise was performed on a treadmill using the Bruce protocol, and the target exercise intensity was maintained by adjusting the speed while maintaining a constant treadmill incline. The tests were performed in a controlled environment at room temperature (23~24 °C) and relative humidity (50~55%). Meanwhile, the SSD group was given a sleep supplement for 30 min after the SD period. They slept on a bed in the laboratory. To create a conducive sleeping environment, the laboratory lights were turned off, and a black curtain was used to block out outdoor lights. Additionally, the temperature and humidity in the laboratory were maintained at 24~25 °C and 40~50%.

### 2.6. Analysis of Oxidative Stress, hs-CRP and Cortisol

To analyze oxidative stress, hs-CRP and cortisol levels, 10 mL of blood was collected from the median cubital vein of participants at three different time points: at rest after NS, at rest after SD and immediately after treatment by group (AT). After allowing the blood samples to clot for 20 min at room temperature, the serum was separated by centrifugation at 3000 rpm for 15 min at room temperature using a centrifuge (Union 32R, Hanil Co., Inchun, Korea). The collected serum was then stored immediately at –80 °C until analysis. 

Serum was used to measure markers of oxidative stress [reactive oxygen metabolites (d-ROMs) and biological antioxidant potential (BAP)] using the free radical analytical system (FRAS4 evolvo; H&D srl, Parma, Italy). The measurement of d-ROMs was performed using the d-ROMs kits (IKIT100 d-ROMs, H&D srl, Parma, Italy). Measurement of BAP was performed with BAP kits (IKIT100 BAP, H&D srl, Parma, Italy). In addition, hs-CRP and cortisol were measured using an immunoassay analysis system (i-CHROMATM reader, Boditech Med Inc., Chuncheon, Korea). Measurement of hs-CRP was performed using hs-CRP assay kits (CFPC-6, Boditech Med Inc., Chuncheon, Korea), and cortisol was measured using cortisol assay kits (CFPC-24, Boditech Med Inc., Chuncheon, Korea). 

### 2.7. Statistical Analysis 

The statistical analysis of this study was performed using SPSS software (version 25.0, IBM Corp, Armonk, NY, USA). Normal distribution of all data was confirmed through the Shapiro-Wilk test, and data are presented as mean ± standard deviation. 

To evaluate significant differences in participants’ baseline characteristics and exercise-related factors among treatment groups, a one-way analysis of variance (ANOVA) was performed, followed by Tukey’s post-hoc test. For the analysis of sleep-related factors, oxidative stress, hs-CRP and cortisol levels, two-way repeated measures ANOVA was used to investigate significant differences between the treatment groups and time points. If a significant interaction between the treatment group and time points was identified, a one-way repeated measures ANOVA with Bonferroni post-hoc test was used to compare differences among time points within each group separately. Additionally, a one-way ANOVA with Bonferroni post-hoc test was used to identify differences among treatment groups at each time point. The statistical significance level (α) was set at 0.05 for all analyses. 

## 3. Results

### 3.1. Participant Characteristics

Table 1 shows the baseline characteristics of the variables for each experimental group. No significant differences were found among groups for any of the factors (*p* > 0.05). Therefore, it identified that the baseline characteristics of groups were homogeneous. 

### 3.2. The Differences in Sleep-Related Factors and Exercise-Related Factors among Groups 

Data are expressed as the mean ± standard deviation. The results of the statistical analysis on the differences in sleep-related factors between the treatment group and time are presented in Table 2. The analysis showed significant differences only at the main effect time point for ATST, ESS and VAS (*p* < 0.001). The post-hoc analysis revealed that ATST significantly decreased in the SD compared to the NS (*p* < 0.001), while ESS and VAS significantly increased in the SD compared to the NS (*p* < 0.001). However, there were no significant differences in proportion of REM sleep, proportion of light sleep, proportion of deep sleep and calorie consumption per day.

Table 3 shows the differences in exercise-related factors among the exercise groups. The results of the analysis indicate that all exercise-related factors among groups were significantly different (*p* < 0.001). Additionally, it was found that all factors were significantly higher in the order of LES, MES and HES. These findings suggest that the sleep intervention was applied equally across all groups, and the exercise intensity assigned to each group was carried out well. 

### 3.3. The Differences in Oxidative Stress, hs-CRP and Cortisol among Groups

The differences in oxidative stress, hs-CRP and cortisol levels between groups and time are presented in Table 4. The d-ROMs showed a significant interaction effect (*p* < 0.01) and a main effect of time (*p* < 0.001). The post-hoc analysis indicated a significant increase in d-ROMs levels in all groups after SD compared to NS (*p* < 0.001). However, the LES group showed a significant decrease in d-ROMs levels at AT compared to SD (*p* < 0.001), while the HES group showed a significant increase compared to SD (*p* < 0.01). Additionally, the LES and MES groups had significantly lower d-ROMs levels in AT than the HES group (*p* < 0.05). 

Similarly, cortisol levels showed a significant interaction effect (*p* < 0.001) and the main effect of time (*p* < 0.001). Post-hoc analysis revealed a significant increase in cortisol levels after SD compared to NS in all groups (*p* < 0.05). However, in the AT, both the SSD and LES groups showed a significant decrease compared to SD (*p* < 0.05), while the HES group showed a significant increase compared to SD (*p* < 0.05). Moreover, the levels of cortisol in AT were significantly lower in the SSD and LES groups than in the MES and HES groups (*p* < 0.05). 

On the other hand, hs-CRP showed a significant difference in the main effect of time (*p* < 0.01), and the post-hoc analysis indicated a significant increase in all groups after SD compared to NS (*p* < 0.05). In contrast, BAP did not exhibit any significant difference in either the main or interaction effect (*p* > 0.05).

## 4. Discussion

This study aimed to investigate the effects of aerobic exercise on oxidative stress, hs-CRP and cortisol levels induced by SD. Through this investigation, the study aimed to confirm the safety and effectiveness of aerobic exercise under SD conditions. 

Sleep duration and quality are crucial factors for maintaining good health in humans. To assess participants’ sleep status, this study used a wearable activity tracker and a self-reported questionnaire to measure sleep duration and quality. The PSQI is typically used as a tool to evaluate sleep quality [25], while the ESS measures daytime sleepiness [26] and the VAS assesses the current level of fatigue [27]. The study found a significant decrease in ATST after SD compared to NS, while the ESS and VAS scores significantly increased after SD compared to NS. These findings are consistent with previous studies that have shown significant impacts of SD on sleep quantity and quality [28,29,30]. Regarding the ESS, a score of less than 8 indicates no daytime sleepiness, while a score of 9 or more indicates presence of daytime sleepiness [31]. Daytime sleepiness is directly related to nighttime sleep duration, and it increases when there is complete or partial sleep deprivation [32]. In this study, all groups had an ESS score of 9 or higher after three days of SD, indicating the presence of daytime sleepiness due to lack of sleep. A higher VAS score indicates greater fatigue [33]. In this study, the VAS score also increased more in SD than in NS, indicating that fatigue felt by individuals increased due to lack of sleep. Therefore, three days of SD in this study caused daytime sleepiness and fatigue during the day, indicating that three days of SD was well induced. 

In this study, we found no significant difference in the daily calorie consumption between the NS period and SD period, indicating that the participants’ physical activity levels remained consistent across both periods. Furthermore, to control for the possible impact of diet on the dependent variables, we provided the same number of calories in the lunch boxes provided to the participants during the experiment period. These measures were taken to ensure that any confounding factors that could affect the dependent variables were minimized and controlled for in this study. 

Oxidative stress occurs when the production of reactive oxygen species (ROS) outweighs the endogenous antioxidant capacity. The intrinsic mechanism of inflammation is the generation of ROS, which can lead to a reduction of antioxidants and increased oxidative stress. Oxidative stress can increase inflammation by inducing pro-inflammatory cytokines. Therefore, oxidative stress and inflammation are strongly connected [34]. Oxidative stress plays a crucial role in various human diseases, such as dyslipidemia, diabetes, hypertension, atherosclerosis, metabolic disorders, cardiovascular diseases, cancer and neurodegenerative diseases [35]. In the relationship between sleep and oxidative stress, it is known that sleep reduces oxidative stress by removing oxidants produced during the daytime [36]. In previous studies, it has been shown that SD increases oxidative stress in the brain [37,38] and causes oxidative damage [39,40]. This study found a significant interaction effect between the treatment group and time on the concentration of d-ROMs, which indicates oxidative stress. Specifically, d-ROMs concentrations were significantly higher after SD than NS in all groups. Furthermore, the concentration of d-ROMs after LES was significantly decreased compared to SD, whereas the concentration of d-ROMs after HES was significantly increased compared to SD. In contrast, the concentration of BAP, which represents antioxidant capacity, did not significantly differ between NS and SD. Therefore, this study found that SD for 3 days caused oxidative stress without affecting the concentration of BAP, which represents antioxidant capacity. These findings are consistent with previous studies [37,38,39]. The increase in oxidative stress observed after SD in this study is likely due to an imbalance between oxidative stress and antioxidant defense mechanisms, as the antioxidant defense mechanisms were not upregulated despite the persistent state of stress. Additionally, exposure to light during nighttime wakefulness can suppress the release of melatonin, which has antioxidant capabilities [41]. Prolonged wakefulness and increased metabolic activity can lead to abnormal ROS production, which can alter cell membrane structure and composition and inhibit antioxidant enzyme activity. Chronic exposure to oxidative stress can impair antioxidant defense mechanisms. Therefore, this study suggests that SD for 3 days acts as a chronic stressor that induces oxidative stress and dysfunction of the antioxidant system. Thus, adequate sleep is crucial for alleviating oxidative stress and maintaining overall health.

Regarding exercise performance under SD, this study found that low- and high-intensity exercise have different effects on oxidative stress induced by SD. These findings are in agreement with previous studies [38,40,42]. A previous study showed that acute sleep deprivation increased oxidative stress in the hippocampus, cortex and amygdala, while treadmill exercise prevented the sleep deprivation-induced increase in oxidative stress [38]. However, exhaustive exercise following SD increased oxidative stress [40,42]. The effect of exercise on the redox balance is complex, depending on the intensity, duration and training level of the exercise [43]. Since exercise increases oxygen consumption, more ROS is produced during exercise than during rest. Excessive exercise induces an increase in oxidative stress, causing lipid peroxidation, DNA damage and protein oxidation in the human body [44,45]. Additionally, excessive exercise causes tissue damage, which induces inflammatory reactions and secondary reactive oxygen compounds produced by neutrophil activity [46]. However, low- and moderate-intensity exercises have been shown to decrease oxidative stress [47,48]. It has also been reported that the exercise intensity, which can protect free radicals and ROS caused by aerobic exercise through self-mechanism, such as antioxidant enzymes without the supplement of external antioxidants, is less than 70% of VO_2_ max [49]. 

On the other hand, few studies have investigated the effect of napping as an intervention method to reduce oxidative stress induced by SD. The present study investigated the effect of SSD on oxidative stress and found no significant effect. This result suggests that a 30-min nap may not provide sufficient time for the activity of antioxidant enzymes to increase and reduce oxidative stress. Therefore, LES is an effective intervention for reducing oxidative stress caused by SD, while HES has a negative effect that may even exacerbate oxidative stress. However, MES and SSD have no impact on changes in oxidative stress. 

Exposure to stress from various factors can activate both the nervous and endocrine systems, which may impact immune function [50]. Cortisol, a hormone that reflects stress levels, is a type of glucocorticoid that plays a critical role in maintaining body homeostasis [51]. Continuous exposure to high levels of cortisol due to stress can lead to various pathological conditions, such as weight loss, hyperglycemia, hypertension and the inhibition of immune function [52]. SD and sleep disorders are known to induce stress in the body. In this study, the concentration of cortisol showed an interaction effect between the group and time. Specifically, cortisol concentrations were significantly higher after SD than NS in all groups. Furthermore, the concentrations of cortisol after both LES and SSD were significantly decreased compared to SD, while the concentration of cortisol after HES was significantly increased compared to SD. In contrast, the concentration of cortisol after MES did not significantly differ from that after SD. 

This finding is consistent with previous studies that have reported an increase in cortisol levels following SD [33,53,54,55]. SD activates the HPA axis, resulting in the release of cortisol. Prolonged SD has been shown to cause adrenal hypertrophy, heightened activity in the sympathetic nervous system (SNS) and release of adrenaline [56]. Sleep restriction can also lead to an increase in the ghrelin hormone, which stimulates the HPA axis and cortisol production [41,57]. Moreover, SD activates the SNS, leading to HPA axis activation, which causes stress and a subsequent increase in cortisol levels. Thus, through these mechanisms, the present study suggests that SD for 3 days triggered stress in the body and resulted in an increase in cortisol levels. 

The relationship between exercise and cortisol has been well-established in previous studies, which have shown that high-intensity exercise leads to an increase in cortisol levels, while low-intensity exercise results in a decrease in cortisol levels [58,59]. The greater the intensity of exercise, the greater the increase in the circulating levels of epinephrine and norepinephrine [60]. In this study, cortisol concentrations decreased with LES and increased with HES, which is consistent with these findings. The intensity of exercise influences the cortisol response of the HPA axis. During high-intensity exercise, the plasma cortisol concentration is high because the secretion rate is faster than the removal rate. Conversely, the decrease in plasma cortisol levels during low-intensity exercise is due to the removal rate being faster than the secretion rate in the adrenal cortex. These changes seem to be due to a combination of hemoconcentration and HPA axis stimulus [58]. The present study also found a reduction in cortisol levels in the SSD, which is consistent with previous studies [61,62]. The reduction in cortisol levels with napping could be attributed to slow-wave sleep inhibiting the HPA axis and cortisol release, as well as the elevated release of catecholamines by the sympathoadrenal system observed following SD [61,62]. Thus, these findings suggest that SSD and LES are effective interventions for reducing stress by removing cortisol caused by SD. However, HES acts as another stressor, which is negative in the body. 

On the other hand, inflammation is a vital immune response that helps maintain tissue homeostasis and combat infection in various harmful conditions [63]. In humans, sleep is known to be closely linked to the immune system, which is a critical defense mechanism in our body. Sleep regulates the function of the immune system and impacts the activation and regulation of inflammatory cytokines [64]. SD or sleep deprivation can have significant adverse effects on the ability to fight infections and alter the intensity and nature of inflammatory responses [65]. Additionally, sleep deprivation can lead to an increase in sympathetic activity, resulting in increased production of pro-inflammatory cytokines, such as IL-1β, IL-6 and TNF-α [66]. 

CRP is a widely used biomarker of inflammation, and it is considered an indicator of the likelihood of cardiovascular disease. Elevated levels of CRP have been independently associated with the progression of atherosclerosis in humans [67,68]. The hs-CRP provides a more sensitive and accurate measurement of CRP at lower concentration levels. In this study, No significant interaction effect between group and time was observed on hs-CRP concentration. However, the concentration of hs-CRP was significantly increased in the SD compared to NS. This finding is consistent with previous studies that have reported an association between SD and increased hs-CRP levels [69,70]. The mechanism through which SD leads to an increase in CRP is related to the activation of a pro-inflammatory signaling pathway that is regulated by toll-like receptors (TLRs) and nuclear factor kappa-beta (NF-κB) [64]. TLRs, one of the innate immune components, are stimulated by SD and induce the production of inflammatory cytokines [71]. Additionally, SD stimulates the activation of NF-kB in the brain, which is related to sleep regulation [72]. 

In the present study, the change in hs-CRP according to the treatment groups after SD was also measured. While all treatment groups showed a tendency to decrease compared to after SD, there was no significant difference. This result is consistent with earlier studies [61,73]. It is known that hs-CRP increases during an inflammatory response and is a non-specific response to cell and tissue metabolism, serving as an increased risk factor for cardiovascular disease prognostic factors [74]. In the present study, although there was no positive effect, such as a reduction in hs-CRP, the absence of an increase in hs-CRP after treatment suggests that the treatment groups are not risk factors for cardiovascular disease. However, further research is necessary to clarify the theory behind these results as the amount of nap and exercise duration may not have been sufficient stimuli to induce hs-CRP changes. 

One of the main limitations of our study is that the induced state of SD was only for three consecutive days, which may not fully reflect the long-term effects of chronic SD. Although our study demonstrated the positive effects of a single session of aerobic exercise on individuals with SD, caution should be exercised in extrapolating these findings to individuals with chronic SD or shift workers. Additionally, the short duration of our SD protocol may not fully reflect the long-term effects of chronic SD. Furthermore, in this study, the presence or absence of snoring, bruxism and sleep abnormalities, such as chronic obstructive sleep apnea, were determined based on the participants’ self-report, which may have limitations. Therefore, future studies should consider using polysomnography to accurately assess the participants’ sleep state during participant selection and experiment. Moreover, the single session of aerobic exercise performed in our study may not accurately reflect the effects of regular exercise on SD over an extended period. Further studies are needed to confirm the beneficial effects of exercise in patients with chronic SD or shift workers, as well as to assess the effectiveness of long-term exercise in reducing the adverse effects of SD.

## 5. Conclusions

This study found that SD increased cortisol and hs-CRP levels, as well as induced oxidative stress. During the SD condition, LES was observed to have a positive effect on reducing oxidative stress and stress hormone levels, whereas HES had negative effects by increasing oxidative stress and stress hormone levels. Therefore, the study suggests that LES is a more suitable exercise intensity for individuals experiencing SD, as it can help reduce oxidative stress and hormone levels that may lead to various health problems. These findings could have significant implications for developing exercise programs for individuals with sleep problems, as well as for the prevention and treatment of related health conditions.

## Figures and Tables

**Figure 1 healthcare-11-01201-f001:**
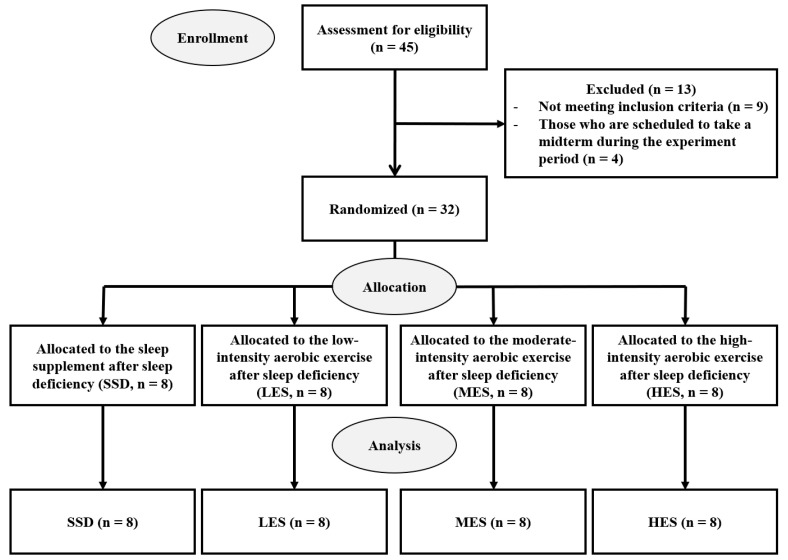
Flow chart.

**Figure 2 healthcare-11-01201-f002:**
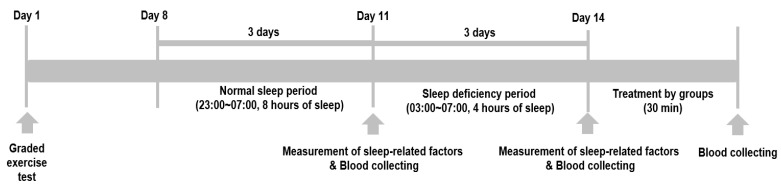
Experimental procedure.

**Table 1 healthcare-11-01201-t001:** The difference in participant baseline characteristics among treatment groups.

Variables	SSD (*n* = 8)	LES (*n* = 8)	MES (*n* = 8)	HES (*n* = 8)	*F*	*p*
Age (years)	20.38 ± 0.52	21.88 ± 2.10	21.38 ± 2.00	21.50 ± 1.77	1.172	0.361
Height (cm)	176.88 ± 4.36	176.88 ± 3.27	174.63 ± 4.47	174.13 ± 5.19	0.887	0.460
Weight (kg)	75.23 ± 8.82	78.68 ± 11.78	70.38 ± 7.13	70.78 ± 5.74	1.657	0.199
BMI (kg/m^2^)	24.03 ± 2.14	25.06 ± 2.95	23.05 ± 1.63	23.36 ± 2.21	1.216	0.322
VO_2_ max (mL/kg/min)	53.33 ± 4.57	52.12 ± 6.77	51.79 ± 5.37	53.29 ± 7.40	0.134	0.939
PSQI (point)	4.13 ± 0.64	3.25 ± 1.28	3.25 ± 0.71	3.25 ± 1.48	1.285	0.299
ESS (point)	4.00 ± 0.93	3.13 ± 1.46	4.25 ± 1.03	3.63 ± 0.92	1.567	0.219
VAS (point)	1.69 ± 0.39	1.75 ± 0.45	1.60 ± 0.61	1.64 ± 0.28	0.167	0.918

Data are expressed as the mean ± standard deviation. Statistical analysis was performed using one-way ANOVA to analyze the differences among treatment groups. Abbreviations: SSD, sleep supplement after sleep deficiency; LES, low-intensity aerobic exercise after sleep deficiency; MES, moderate-intensity aerobic exercise after sleep deficiency; HES, high-intensity aerobic exercise after sleep deficiency; BMI, body mass index; PSQI, Pittsburgh Sleep Quality Index; ESS, Epworth Sleepiness Scale; VAS, Visual Analogue Scale.

**Table 2 healthcare-11-01201-t002:** The differences in sleep-related factors among groups.

Variables		NS	SD	*F*	*p*
ATST (min)	SSD	397.46 ± 7.97	212.88 ± 19.09 ***	G: 2.218 T: 2556.327 G*T: 2.438	0.108 <0.001 0.085
	LES	417.25 ± 10.32	220.42 ± 8.68 ***		
	MES	410.13 ± 17.67	227.38 ± 21.37 ***		
	HES	416.54 ± 25.93	208.17 ± 9.96 ***		
REM (%ATST)	SSD	19.73 ± 2.90	20.91 ± 2.60	G: 0.368 T: 3.425 G*T: 1.699	0.776 0.075 0.190
	LES	19.38 ± 6.24	18.39 ± 1.90		
	MES	20.92 ± 4.51	18.47 ± 2.42		
	HES	20.66 ± 2.86	17.43 ± 4.22		
Light sleep (%ATST)	SSD	62.17 ± 6.06	61.83 ± 6.19	G: 1.059 T: 2.522 G*T: 1.948	0.382 0.123 0.145
	LES	63.58 ± 9.80	66.28 ± 10.90		
	MES	60.82 ± 5.68	59.78 ± 6.46		
	HES	61.15 ± 3.41	68.02 ± 4.91		
Deep sleep (%ATST)	SSD	18.10 ± 4.92	18.51 ± 3.95	G: 0.661 T: 1.296 G*T: 1.685	0.583 0.265 0.193
	LES	17.03 ± 6.34	19.08 ± 6.41		
	MES	18.26 ± 3.82	21.75 ± 4.87		
	HES	18.20 ± 3.23	16.30 ± 3.28		
Calorie consumption per day (kcal)	SSD	2889.56 ± 581.33	2819.92 ± 316.27	G: 2.307 T: 0.136 G*T: 0.866	0.098 0.715 0.471
	LES	3149.38 ± 494.18	3183.29 ± 427.43		
	MES	2802.00 ± 431.44	2745.17 ± 123.59		
	HES	2697.50 ± 156.25	2883.25 ± 314.22		
ESS (points)	SSD	5.00 ± 3.21	11.25 ± 4.83 ***	G: 0.808 T: 90.868 G*T: 2.229	0.500 <0.001 0.107
	LES	3.38 ± 1.92	9.38 ± 3.11 ***		
	MES	5.25 ± 1.91	10.00 ± 4.28 **		
	HES	3.75 ± 1.75	13.38 ± 5.48 ***		
VAS (points)	SSD	4.29 ± 1.76	7.44 ± 1.58 ***	G: 0.741 T: 176.454 G*T: 1.262	0.536 <0.001 0.306
	LES	3.65 ± 1.60	6.99 ± 0.84 ***		
	MES	3.26 ± 1.42	6.96 ± 1.26 ***		
	HES	3.36 ± 1.25	7.91 ± 0.76 ***		

Data are expressed as mean ± standard deviation. Statistical analysis was performed using a two-way repeated measures ANOVA. Abbreviations: NS, normal sleep; SD, sleep deficiency; SSD, sleep supplement after sleep deficiency; LES, low-intensity aerobic exercise after sleep deficiency; MES, moderate-intensity aerobic exercise after sleep deficiency; HES, high-intensity aerobic exercise after sleep deficiency; ATST, actual total sleep time; REM, rapid eye movement; ESS, Epworth sleepiness scale; VAS, visual analogue scale; G, group; T, time; G*T, interaction of group and time. *** denotes a significant difference compared to NS within each group (*p* < 0.001). ** denotes a significant difference compared to NS within each group (*p* < 0.01).

**Table 3 healthcare-11-01201-t003:** The differences in exercise-related factors among groups.

Variables	LES (a)	MES (b)	HES (c)	*F*	*p*	Post-hoc
VO_2_ (mL/kg/min)	21.25 ± 2.74	33.71 ± 2.96	43.31 ± 4.93	72.320	<0.001	a < b < c
HR (beats)	120.60 ± 9.36	145.41 ± 9.28	172.15 ± 2.83	87.783	<0.001	a < b < c
Speed (mph)	2.32 ± 0.32	3.08 ± 0.22	4.16 ± 0.65	35.651	<0.001	a < b < c
Calorie consumption (kcal)	250.25 ± 25.52	352.75 ± 56.53	455.88 ± 48.32	41.033	<0.001	a < b < c
Running distance (m)	2297.25 ± 250.89	2912.25 ± 142.36	3651.38 ± 457.08	37.766	<0.001	a < b < c

Data are expressed as the mean ± standard deviation. Statistical analysis was performed using one-way ANOVA. Abbreviations: LES, low-intensity aerobic exercise after sleep deficiency; MES, moderate-intensity aerobic exercise after sleep deficiency; HES, high-intensity aerobic exercise after sleep deficiency; HR, heart rate.

**Table 4 healthcare-11-01201-t004:** The differences in oxidative stress, hs-CRP and cortisol between groups and times.

Variables		NS	SD	AT	*F*	*p*
d-ROMs (U.Carr)	SSD	279.25 ± 24.01	342.75 ± 15.59 ***	341.00 ± 16.54 ***	G: 1.476 T: 215.466 G*T: 3.735	0.242 <0.001 0.003
	LES	262.88 ± 11.19	344.25 ± 11.47 ***	313.88 ± 26.46 ***^###a^		
	MES	262.63 ± 20.15	342.13 ± 28.92 ***	332.00 ± 17.14 ***^a^		
	HES	274.00 ± 24.28	338.00 ± 24.05 ***	356.00 ± 26.55 ***^#^		
BAP (μmol/L)	SSD	2414.50 ± 178.93	2426.13 ± 174.40	2431.88 ± 119.59	G: 1.609 T: 1.301 G*T: 0.580	0.210 0.217 0.580
	LES	2473.25 ± 121.38	2584.50 ± 77.97	2562.63 ± 59.85		
	MES	2497.00 ± 169.42	2491.63 ± 181.33	2481.25 ± 130.23		
	HES	2408.88 ± 159.52	2471.38 ± 181.33	2484.50 ± 100.26		
hs-CRP (mg/L)	SSD	0.36 ± 0.31	0.52 ± 0.47 *	0.42 ± 0.39	G: 0.923 T: 8.458 G*T: 0.433	0.442 0.003 0.780
	LES	0.30 ± 0.13	0.46 ± 0.25 *	0.45 ± 0.24		
	MES	0.43 ± 0.24	0.75 ± 0.69 *	0.71 ± 0.66		
	HES	0.53 ± 0.43	0.76 ± 0.47 *	0.69 ± 0.51		
cortisol (nmol/L)	SSD	418.26 ± 78.94	478.20 ± 92.46 *	330.05 ± 86.32 *^#ab^	G: 1.471 T: 10.826 G*T: 6.605	0.244 <0.001 <0.001
	LES	371.57 ± 81.70	441.61 ± 99.08 *	364.77 ± 47.76 ^#ab^		
	MES	364.02 ± 62.19	446.65 ± 103.21 *	448.20 ± 97.23 *		
	HES	393.50 ± 75.85	455.54 ± 45.63 *	516.90 ± 37.71 *^#^		

Data are expressed as the mean ± standard deviation. Statistical analysis was performed using a two-way repeated measures ANOVA. Abbreviations: NS, normal sleep; SD, sleep deficiency; AT, immediately after treatment by group; SSD, sleep supplement after sleep deficiency; LES, low-intensity aerobic exercise after sleep deficiency; MES, moderate-intensity aerobic exercise after sleep deficiency; HES, high-intensity aerobic exercise after sleep deficiency; d-ROMS, reactive oxygen metabolites; BAP, biological antioxidant potential; hs-CRP, high sensitive C-reactive protein; G, group; T, Time; G*T, interaction of group and time. * denotes a significant difference compared to NS within each group (*p* < 0.05). *** denotes a significant difference compared to NS within each group (*p* < 0.001). ^#^ denotes a significant difference compared to SD within each group (*p* < 0.05). ^###^ denotes a significant difference compared to SD within each group (*p* < 0.001). ^a^ denotes a significant difference compared to HES at the AT (*p* < 0.05). ^b^ denotes a significant difference compared to MES at the AT (*p* < 0.05).

## Data Availability

The data presented in the study can be available from the corresponding author upon reasonable request.

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
