# Peer review of "The Effect of Aerobic Exercise on Variation of Oxidative Stress, hs-CRP and Cortisol Induced by Sleep Deficiency"

_healthcare, 2023, doi:10.3390/healthcare11081201_

Round 1

Reviewer 1 Report

Excellent work. It was a pleasure to review this manuscript. Study design is sound, results are not overinterpreted, and limitations are acknowledged appropriately. Such a relief to review a good manuscript.

The only error I found was using the term "measures" in line 338. Cortisol "reflects" stress levels, or "responds to" stress levels. The original statement should be reworded.

Well done!

Reviewer 2 Report

Dear authors,

I carefully read your manuscript aiming to assess the effects of different physical activity intensities on young people undergoing partial sleep deprivation.

Some passages, especially in the study protocol,  are unclear, making understanding the results very difficult.

Please, find my comments below:

INTRODUCTION:

1.       In general, most of your statements (even an entire paragraph) are completely unreferenced. I suggest supporting your statements through proper references.

2.       In lines 46-53, I suggest adding 10.1139/H07-018 10.3390/nu12123622

3.       Line 34, what do you mean by high-quality life?

4.       Lines 33-34, sleep is also necessary for physical performances (doi: 10.1016/j.physbeh.2022.113963, doi: 10.1016/j.ncl.2017.03.002)

5.       It is not clear (since there are insufficient references) how many studies focused on SD and physical activity in the general population. It seems they are very few; however, reading lines 100 (referring) and 267 (confirm), it seems the topic has already been treated.

METHODS

6.       The study protocol is not clear

a.       How much before the NS period has been performed the tests; thus, how much time before the start of the DS period?

b.       When did participants perform the tests after the DS period?

c.       Did the aerobic performance take place every day at the same time?

7.       Did you obtain only ATST from Fitbit? Since yours is a sleep deprivation study, it is evident that the sleep amount decreased. It would be interesting seeing the sleep efficiency, immobile minutes, sleep fragmentation, and other sleep parameters assessing sleep quality. These parameters could also be evaluated concerning physical activity intensities and inflammation parameters.

8.       All the questionnaires are missing the evaluated time.

9.       The PSQI evaluates sleep during the last 30 days. How much time elapses between the first and the second administration? This questionnaire could not be the most suitable for your study protocol; you should have evaluated an overlying time. It would be best if you addressed this.

RESULTS

10.   It is obvious and useless to report that ATST decreased and calories were superimposable (you controlled meals).

11.   Table 4, what do you mean by AT? When were these data collected? That is why the study protocol needs to be more apparent.

12.   Data for SSD are never reported? Why did you not compare the four groups, But only the three undergoing physical activity interventions?

DISCUSSION

13.   Line 268, how did you assess safety?

14.   Discussion centres entirely on inflammation results, whereas questionnaire results are minimally discussed. Why? Are they useful for the results, or could they be omitted?

15.   You always said all groups without considering that the SSD is not included.

Reviewer 3 Report

Thank you for the possibility to review manuscript titled: The Effect of Aerobic Exercise on Variation of Oxidative Stress, hs-CRP, and Cortisol Induced by Sleep Deficiency I found a lot of strength of this: it is well written, the topic is interesting both for clinicians and researchers. However there are also flaws and shortcomings:   Major issues:  The main issue is the flawed methodology. No polysomnography (psg) was  performed, therefore authors could not exclude sleep disorders. Obstructive sleep apnea is not recognized commonly, therefore psg or at least questionnaire (ie STOP-Bang) should be conducted. Moreover, in young subject in their twenties, sleep bruxism is prevalent and cause sleep fragmentation. It also could influence the results of the study (ESS, fatigue scale, sleep quality etc) Authors research the consequences of sleep deprivation. However they did not assess properly hours and quality of sleep. The polysomnography should be conducted, which is gold standard. There is no information where the patient slept the first three nights; probably in home, because text message were sent to them. It is a serious limitation also, because home environment is not suitable for such sleep test. Noises from other household members or pets often disturb sleep. The  effect of light  was also not controlled. Thus the test should be performed in sleep laboratory. What is more, the authors stated that “researchers supervised the participants by using mobile messages to ensure that they adhered to the experimental protocols” Thus subject used their smartphones before sleep which is serious limitation because of  light exposure  from smartphones. The authors rightly note in discussion section that their study results are consistent with earlier ones. Regrettably I did not find any novel result. Minor issues: 1)Abstract is difficult to follow because the abbreviations are not expanded(ie d-ROM) 2) The aim should not be described in conclusion section . 3) How many subjects were enrolled before exclusions? How many were excluded? Flowchart of participants recruitment would be useful.

Reviewer 4 Report

Manuscript measures impact of short-term (3 days) of reduced sleep on various measures of oxidative stress and immune/stress responses. It then tests the impact of sleep supplement and several intensities of exercise in protecting against these effects. This is a clearly designed study with some limitations (acknowledged by the authors in discussion). Nevertheless, the study clearly replicates several known effects of reduced sleep even in the short time frame, and gives a clear indication that exercise, especially lower-intensity exercise has measurable protective effects against accumulated stress responses. A couple of minor additions would increase the value of the manuscript, which is already quite nice.

1. Legends for Tables 1 and 2 should clearly define the terms G and T as group and time. Readers may be able to infer this, but it should be stated in the legend, as it is in later tables. 

2. Would be worth addressing in the discussion potential mechanisms that may explain why higher intensity exercise was less effective. Could it have to do with the ability of higher intensity exercise to further stimulate adrenergic signaling, which is already increased during SD? Authors may be able to gain some insight into this from literature that measures norepinephrine release during higher intensity exercise and comparing this to HR and VO2 measurements in this study? 

Round 2

Reviewer 2 Report

Dear authors,

The manuscript improved after the revision; however, the study protocol is still unclear, and I still have some concerns:

1. NS must be abbreviated in the text, not only in the tables' footnotes.

2. Figure 2 makes the study protocol clearer; however, it still needs to be better explained in the text:

2.a how many days have elapsed between day 14 and the last blood collection?

2.b what does "treatment group (30 minutes) mean?

2c. did the exercise group train every day or when? Is the test performed only at the end of the protocol? In the last case, did you expect that undefined minutes of an aerobic test could have nullified or modified the oxidative stress or inflammation consequences of 3 SD nights?

3. probably, my previous comment 6 has been misunderstood: you must make explicit in lines 148-163 that the PSQI usually evaluates sleep in the past month, VAS usually assess fatigue in the past (I do not know)..., and ESS usually considers sleepiness in the past (I do not know), and that you adapted these periods to your study protocol.

4. how did you obtain results in table 2? with a two-way repeated ANOVA? If so, this is reported nowhere in the text.

5. NS must be abbreviated in the text, not only in the tables' footnotes.

6. lines 289-291: you must explain what an increase in PSQI, ESS and VAS means, not only say that they increased. Moreover, since you wrote only two lines about questionnaires in the discussion and considering that the PSQI has been misused, is it necessary to report the results of the questionnaire, or could they be avoided?

7. line 292: how did you control for potential confounding factors in your analysts, since you did not use covariates and physical activity was an effect rather than a founding effect?

8. did you check for the physical activity habits and training at enrollment? Did you reflect that different physical activity levels could have differently affected the results? A well-trained person could react differently in d-ROMS and CPR than a less-trained person. Moreover, a high-intensity, also a single bout at that intensity, exercise could increase d-ROMs and inflammation rather than decrease them. Thus, the changes you see in table 4 could be traceable in exercise consequences rather than sleep ones. However, it is challenging to make some considerations due to the unclear study protocol.

Reviewer 3 Report

Authors answer the questions and notes of reviewer , however manuscript has not been improved. Authors agree that methodology is flawed .Regrettably they did not discuss it in limitation section. Summarizing , because of flawed methodology, small sample size, lack of novelty, I do recommend to reject the manuscript.
